# The Extended Multidimensional Neo-Fuzzy System and Its Fast Learning in Pattern Recognition Tasks †

**Yevgeniy Bodyanskiy [1], Nonna Kulishova [2,* and Olha Chala [1]**

[1]   Control Systems Research Laboratory, Kharkiv National University of Radioelectronics, 61166 Kharkiv, Ukraine; yevgeniy.bodyanskiy@nure.ua (Y.B.); olha.chala@nure.ua (O.C.)

[2]   Media Systems and Technologies Department, Kharkiv National University of Radioelectronics, 61166 Kharkiv, Ukraine

*   Correspondence: nokuliaux@gmail.com; Tel.: +38-67-257-9275

†   This paper is an extended version of our paper: Bodyanskiy, Y., Kulishova, N., Malysheva, D. The Multidimensional Extended Neo-Fuzzy System and its Fast Learning for Emotions Online Recognition. In Proceedings of the 2018 IEEE Second International Conference on Data Stream Mining and Processing (DSMP), Lviv, Ukraine, 21–25 August 2018, pp. 473–477.

**Abstract:** Methods of machine learning and data mining are becoming the cornerstone in information technologies with real-time image and video recognition methods getting more and more attention. While computational system architectures are getting larger and more complex, their learning methods call for changes, as training datasets often reach tens and hundreds of thousands of samples, therefore increasing the learning time of such systems. It is possible to reduce computational costs by tuning the system structure to allow fast, high accuracy learning algorithms to be applied. This paper proposes a system based on extended multidimensional neo-fuzzy units and its learning algorithm designed for data streams processing tasks. The proposed learning algorithm, based on the information entropy criterion, has significantly improved the system approximating capabilities. Experiments have confirmed the efficiency of the proposed system in solving real-time video stream recognition tasks.

**Keywords:** extended multidimensional neo-fuzzy system; pattern recognition; entropy-information criterion; extended neo-fuzzy units

## 1. Introduction

Classification and clustering relate to the main tasks in data stream analysis. They mean the distribution of objects between groups with not known properties in advance. In video analytics, there is a task that arises often; when it needs to find the specified objects the video sequence and track their movement in the frame. Another problem is the object state changes detection, which remains within the frame for a long time. Video streams are characterized by a non-Gaussian nature. Many methods of clustering and classification have been developed for their analysis. Among these methods, a large group consists of methods based on the use of ANNs. A classic approach can be found in [1]. Modern classification and clustering systems often use SVM, which provide high accuracy [2–5]. Solutions were also obtained for weighted fuzzy support vector regression [6], radial basis networks [7].

Specifically, Deep Neural Networks (DNN) [8–11] have been given in terms of classification accuracy. This line of research turned out to be very promising, convolutional neural networks (CNN) and deep learning are widely used in the pattern recognition tasks, especially for image, audio and video streams classifying and clustering. For recognition of Chinese characters, a convolutional network with the RNN Framework is proposed in [12]. The Levenberg–Marquardt network [13] is used

in the diagnostic task for medical photos. Deep recurrent architecture was at the heart of the system for remote sensing image classification [14]. However, DNNs are not prone to certain shortcomings, with a low speed of multiepoch learning being the major limitation, leading to their inefficiency in a subclass Data Stream Mining tasks, when information is fed to the system in a sequential mode (quite large volume of training sets that are not always available). Furthermore, in real-world tasks, image clusters are often mutually overlapped, which calls for L. Zadeh's fuzzy logic classification methods. In this case, hybrid systems of computational intelligence [15,16] come in handy, since they both possess the learning capabilities of ANN and DNN and, being a fuzzy systems, are capable of distinguishing overlapping classes. The training speed of such systems calls for attention which requires the use of non-standard neurons, architectures and teaching methods.

In [17], the mixed fuzzy clustering algorithm for health care problems where time series analysis is necessary was proposed. Another hybrid structure discussed in [18] is the deep TSK classifier which uses interpreted linguistic rules for a fuzzy inference system.

The high training speed of the hybrid systems requires the use of non-standard neurons, architectures and teaching methods.

In this paper, we consider an approach to constructing a hybrid system that performs the multidimensional data classification using neo-fuzzy neurons. The structure was developed for the problem of emotion estimation by images and video streams. Therefore, Section 2 describes previous works that are relevant to this task. Section 3 describes in detail the architecture of classification system. Section 4 is devoted to the learning algorithm of the neo-fuzzy structure. Section 5 discusses the results of experiments, and the conclusions are presented in Section 6.

## 2. Related Work

Many important practical tasks in the video analytics, such as health care and life support, crowd analytics, surveillance, man-machine interface and so on, are now associated with systems capable to recognize the emotional status [19]. Researches in this area are connected with primary data gathering methods on video and audio streams [20,21] and with methods for human emotions online classification, clustering and recognition [22–24]. Artificial neural networks are particularly effective in analyzing nonlinear processes in real-time; so, many researchers use them to identify human facial expressions. In [25,26], the use of genetic algorithms is considered; in [27] the use of cascaded continuous regression; in [28] the use of shallow neural networks. Reference [6] describes a fuzzy system for emotional intent classifying, while [29] describes the affect estimation by audio stream using ensemble of ordinal classifiers. Many works have been devoted to the recognition of emotions from photos and videos using deep CNN, for example [30,31].

The proposed system is based on the neo-fuzzy approach [32–34], which provides high approximating properties and, therefore, can be applied in solving a number of real practical tasks [35,36]. It is also important to note that a neo-fuzzy system learning rate can be optimized [37], which allows using it in real-time Data Stream Mining tasks.

## 3. The Architecture of the Neo-Fuzzy Classification System

Neo-fuzzy neuron modifications, such as extended neo-fuzzy neurons (ENFN) [38–40] and neo-fuzzy units (NFU) [41–43] with nonlinear activation (sigmoidal) functions have significantly improved approximating capabilities. In the system proposed here, we suggest utilizing a hybrid neo-fuzzy unit, which is a modification of an NFU which replaces the standard nonlinear synapses (NS) with extended nonlinear synapses (ENS) taking advantage of the neuro-fuzzy Takagi-Sugeno–Kang arbitrary order system properties.

Figure 1 shows the proposed extended multidimensional neo-fuzzy system (EMNFS) architecture with two information processing layers.

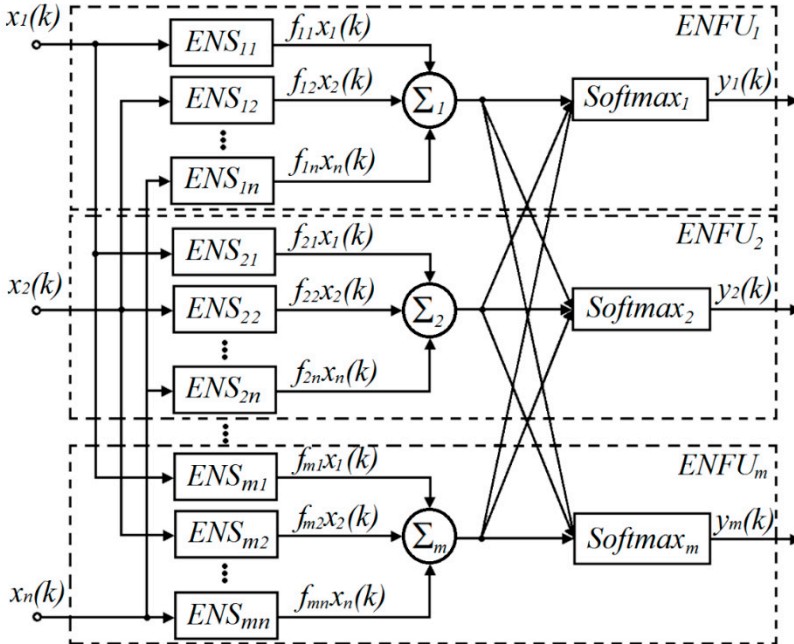

**Figure 1.** The architecture of the proposed extended multidimensional neo-fuzzy system (EMNFS).

The vector signal $x(k) = (x_1(k), x_2(k), \ldots, x_i(k), \ldots, x_n(k))^T \in R^n$ of images to be classified is fed to the inputs of the system, where $k = 1, 2, \ldots$ is the index of the current discrete time. The first layer of the system is formed by extended nonlinear synapses (ENS), where $j = 1, 2, \ldots, m$, m is the number of possible classes. For each $j$-th class, n such synapses $ENS_{j1}$, $ENS_{j2}$,..., $ENS_{jn}$, whose output signals $f_{j1}(x_1(k)), f_{j2}(x_2(k)), \ldots, f_{jn}(x_n(k))$ are fed to the summation blocks $\Sigma_j$, are used. $ENS_{ji}$ synapses and adders $\Sigma_j$ form extended neo-fuzzy neurons (ENFN) [38,39]. ENFN output signals are fed to the second output layer of the system, formed by nonlinear softmax activation functions:

$$soft\text{max} u_j = \frac{\exp(u_j)}{\sum\limits_{j=1}^{m} \exp(u_j)}, \tag{1}$$

which are the generalization of traditional sigmoidal activation functions

$$\sigma(u_j) = \frac{1}{1 + \exp(-u_j)}$$

for classification systems with many outputs [10]. $ENFN_j$ together with functions $soft\text{max} u_j$ form extended neo-fuzzy units (ENFU), which are a generalization of neo-fuzzy units introduced in [40,41]. Output signals of the system

$$y_j(k) = soft\text{max}(u_j(k)), \ \sum_{j=1}^{m} y_j(k) = 1 \tag{2}$$

specify the levels of presented input image $x(k)$ fuzzy membership to each of the possible m classes.

As is known [32–34], the standard neo-fuzzy neuron is formed by n nonlinear synapses; each of them implements the fuzzy derivation of the Takagi–Sugeno–Kang of zero order (Wang–Mendel reasoning)

$$\text{if } x_i \text{ is } X_{li} \text{ then } \varphi_i(x_i) \text{ is } w_{li}, l = 1, 2, \ldots, h$$
$$f(x_i) = \sum_{l=1}^{h} \varphi_{li}(x_i) = \sum_{l=1}^{h} w_{li}\mu_{li}(x_i),$$

where h is the number of membership functions $\mu_{li}(x_i)$ in the nonlinear synapse, $w_{li}$ adjustable synaptic weights, $i = 1, 2, \ldots, n$ synapse number in the neo-fuzzy neuron.

The extended nonlinear synapse introduced in [38,39] implements the Takagi–Sugeno–Kang inference of arbitrary order, that is:

$$\text{if } x_i \text{ is } X_{li} \text{ then } \varphi_i(x_i) \text{ is } w_{li}^0 + w_{li}^1 x_i + \ldots + w_{li}^p x_i^p, l = 1, 2, \ldots, h$$

$$f_i(x_i) = \sum_{l=1}^{h} \varphi_{li}(x_i) = \sum_{l=1}^{h} \mu_{li}(x_i)\left(w_{li}^0 + w_{li}^1 x_i + w_{li}^2 x_i^2 + \ldots + w_{li}^p x_i^p\right) =$$
$$= \sum_{l=1}^{h} w_{li}^0 \mu_{li}(x_i) + w_{li}^1 x_i \mu_{li}(x_i) + w_{li}^2 x_i^2 \mu_{li}(x_i) + \ldots + w_{li}^p x_i^p \mu_{li}(x_i).$$

By introducing for $\text{ENS}_{ji}$ a vector of synaptic weights $w_{ji} = \left(w_{j1i}^0, w_{j1i}^1, w_{j1i}^2, \ldots, w_{j1i}^p, \ldots, w_{jli}^0, \ldots, w_{jli}^p \ldots, w_{jhi}^p\right)^T$ and fuzzyficated signals $\tilde{\mu}_{ji}(x_i) = \left(\mu_{j1i}(x_i), x_i\mu_{j1i}(x_i), \ldots x_i^p \mu_{j1i}(x_i), \ldots, \mu_{jli}(x_i), x_i\mu_{jli}(x_i), \ldots x_i^p \mu_{jli}(x_i), \ldots, x_i^p \mu_{jhi}(x_i)\right)^T$ of dimensionality $(p+1)h \times 1$, we can write the output of this synapse in the form

$$f_{ji}(x_i) = w_{ji}^T \tilde{\mu}_{ji}(x_i).$$

Further, it is easy to write the output of each $\text{ENFN}_j$ as a whole in the form

$$u_j(k) = \sum_{i=1}^{n} f_{ji}\left(x_i^{(k)}\right) = \sum_{i=1}^{n} w_{ji}^T \tilde{\mu}_{ji}\left(x_i^{(k)}\right).$$

Introducing the vectors of $\text{ENFN}_j$ synaptic weights $w_j = \left(w_{j1}^T, w_{j1}^T, \ldots, w_{jn}^T\right)^T$ and fuzzyficated inputs $\tilde{\mu}_j(x) = \left(\tilde{\mu}_{j1}^T(x_1), \ldots, \tilde{\mu}_{ji}^T(x_i), \ldots, \tilde{\mu}_{jn}^T(x_n)\right)^T$ of the $(p+1)h \times n$ dimensionality, the output signal $\text{ENFN}_j$ can be rewritten in compact form:

$$u_j(k) = w_j^T \tilde{\mu}_j\left(x^{(k)}\right) \quad \forall j = 1, 2, \ldots, m.$$

This signal is then fed to the activation function of the output layer, and a whole resulting system output form the values of the fuzzy membership levels of the presented image $x(k)$ to the $j$-th class:

$$y_j(k) = soft\text{max}\left(w_j^T(k-1)\tilde{\mu}_j\left(x^{(k)}\right)\right),$$

where $w_j(k-1)$ are the values of the synaptic weights obtained as a result of learning on previous $(k-1)$ images.

## 4. Learning of the Extended Multidimensional Neo-Fuzzy System in the Pattern Recognition Task

To learn the considered system, it is advisable to use the cross-entropy learning criterion [1,33]

$$E(k) = -\sum_{j=1}^{m} d_j(k) \ln y_j(k) \tag{3}$$

and the one-hot coding of the reference signal, when the vector external reference signal $d(k) = \left(d_1(k), \ldots, d_j(k), \ldots, d_m(k)\right)^T$ is formed by zeros and a single unit located in a position corresponding to the "correct" class.

Minimizing the criterion (3) with the standard gradient procedure leads to the δ-rule setting of the synaptic weights of each ENFU$_j$ in the form

$$w_j(k) = w_j(k-1) - \eta_j(k)\nabla_{w_j}E(k) = w_j(k-1) + \eta_j(k)e(k)\widetilde{\mu}_j(x(k)) \tag{4}$$

where $\eta_j(k)$ is the learning rate parameter of $j$-th ENFU, $e(k) = d(k) - y_j(k)$ a learning error, while the $j$-th component of vector reference signal $d(k)$ can take only two values—0 or 1.

Algorithm (4) can be given both filtering and tracking properties, if the parameter $\eta_j(k)$ is chosen in accordance with the relation [37]

$$\eta_j^{-1}(k) = r_j(k) = \alpha r_j(k-1) + \|\widetilde{\mu}_j(x(k))\|^2, \tag{5}$$

where $0 \leq \alpha \leq 1$ is the forgetting factor. Depending on the value of $\alpha$, procedures (4),(5) can take stochastic approximation properties for $\alpha = 1$ (Goodwin–Ramadge–Caines algorithm) [42] or for $\alpha = 0$ the procedure takes the form of the optimal Kaczmarz–Widrow–Hoff algorithm [43], which provides the maximal speed of convergence to optimal solution.

## 5. Experiments

Among the areas where real-time recognition results are extremely important, we highlight human–computer interfaces of various kinds. In many situations, the psycho-emotional state of one person, for example, wakefulness of vehicles drivers and nuclear objects operators are crucial. Just as in the care of seriously ill or lonely patients, it is sometimes necessary to carry out continuous monitoring of their condition and timely detection of deviations.

The problem of emotional status recognition is further complicated by the fact that people often perceive not one, but a whole range of emotions. In this range, individual emotions can be expressed in varying degrees, forming a certain combination. In communication, people "read" these degrees of individual emotion expression, and, by their specific weight, they form an idea of the state of the interlocutor. Thus, the basic emotions can be represented in the form of fuzzy variables, which are expressed by their membership value, which lies within the limits [0; 1]. Obviously, the automatic recognition system should then produce the appropriate output signal. The advantage of the approach proposed in this paper is that the output layer of the extended multidimensional neo-fuzzy system forms a vector whose elements are in the desired range [1].

To study the designed architecture and learning algorithm an experiment on basic emotions recognition was performed. The images from PICS and CK+ open databases [44,45] were used as the objects for recognition. PICS image database consists of single images of individuals in different emotional states. The CK+ database contains separate frames from video sequences with transitions between different emotional states of dozens of people. In all the photos, people are photographed from the front, without tilting the head, in standard lighting, with the same distance from the camera. 87 photos were taken from the base of the CK+, 257 from the PICS base; on the selected images, there are reflections of pure emotions only.

Using some contour detectors like SURF, BRISK or Shi-Tomasi [46–48], we had obtain feature points for every selected image. Input data vector contain X and Y coordinates of 35 features, including such points like, for example:

- eyebrows centers and corners;
- eye centers;
- nose end;
- nose wings;
- mouth center;
- corners of the mouth;
- lips centers;

- nasolabial folds;
- earlobes;
- chin;
- contours of lower jaw and so on.

The point's placement can indicate the basic facial actions of the FACS system in the facial dynamics (Figure 2). Chosen feature points are connected with Facial action units and allow recognizing investigated emotions.

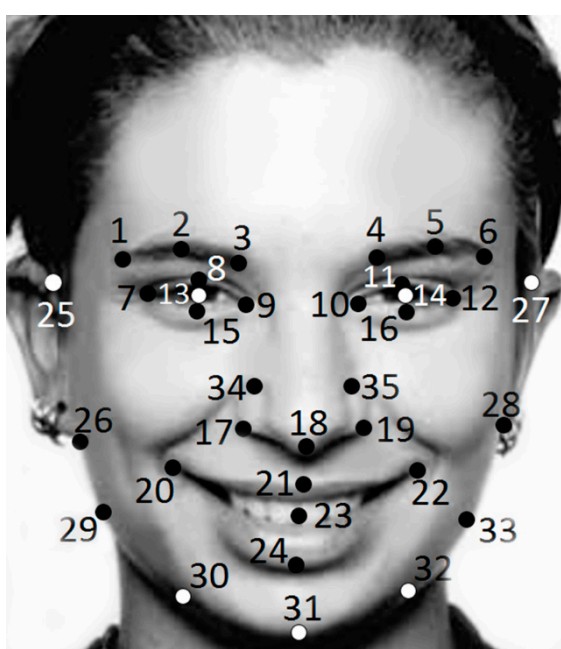

**Figure 2.** Facial features placement for emotion recognition.

The output vector corresponds to a set of seven simplest emotions (neutral, exasperation; distaste; anxiety; grief; astonishment; joy). The number of neo-fuzzy neurons in the first layer varying from 3 to 11.

The learning data contains 344 images, learning repeated from 10 to 10,000 epochs. This paper deals with the case when the data for learning is small. To assess whether the proposed architecture is able to recognize facial expressions, small sets of photos are used. Each set contained no more than 100 pictures.

The network was being learned to recognize emotions by a set of photos grouped for one given facial expression. Then, the system was put into fuzzy-reasoning mode. Proposed architecture approximating ability was examined on an integrated data set, where photos with different emotions were mixed, and their total number was 344. The learning set consists of 60% of all photos, selected randomly from 344. To test the model used the remaining 40%. The experiments were carried out more than 30 times; before each run, the initial data was smashed so that in different experiments the learning and testing data did not coincide. The final results of the experiments are quite close to each other, and the difference did not exceed 2–3%.

The network learning error is shown in Table 1 as number of unrecognized emotions depending on the number of learning epochs and the number of neo-fuzzy neurons. The number of membership functions in every nonlinear synapse in neo-fuzzy neuron was 3.

**Table 1.** Data set size and number of unrecognized images.

| | Emotions | | | | | | |
|---|---|---|---|---|---|---|---|
| | **Exasperation** | **Distaste** | **Anxiety** | **Joy** | **Grief** | **Astonishment** | **Neutral** |
| Data set size | 49 | 66 | 35 | 45 | 19 | 50 | 80 |
| The number of neo-fuzzy neurons in the first layer is 3 | | | | | | | |
| 500 learning epochs | | | | | | | |
| The number of unrecognized emotions in the learning data | 32 | 26 | 30 | 28 | 15 | 18 | 55 |
| 700 learning epochs | | | | | | | |
| The number of unrecognized emotions in the learning data | 22 | 18 | 23 | 22 | 10 | 13 | 41 |
| 1500 learning epochs | | | | | | | |
| The number of unrecognized emotions in the learning data | 15 | 12 | 18 | 15 | 8 | 12 | 32 |
| The number of neo-fuzzy neurons in the first layer is 11 | | | | | | | |
| 5000 learning epochs | | | | | | | |
| The number of unrecognized emotions in the learning data | 0 | 1 | 0 | 1 | 0 | 0 | 1 |
| 7000 learning epochs | | | | | | | |
| The number of unrecognized emotions in the learning data | 0 | 0 | 0 | 1 | 0 | 0 | 0 |
| 10000 learning epochs | | | | | | | |
| The number of unrecognized emotions in the learning data | 0 | 0 | 0 | 0 | 0 | 0 | 0 |

## 6. Conclusions

This article proposes an extended multidimensional system, based on neo-fuzzy neuron modifications, specifically, extended multidimensional neo-fuzzy units, with improved approximating properties.

Experiments have shown that three to five neo-fuzzy neurons in the classification system input layer provides low accuracy, which cannot be enhanced only by learning epochs number increasing. At the same time, an increment of a neo-fuzzy neuron number in the input layer dramatically increases the classification accuracy. The experiments also varied the number of terms in nonlinear synapses from three to seven and changed the Takagi–Sugeno's fuzzy inference order. However, this factor did not have a significant impact on improving the classification accuracy and, therefore, the results of these experiments are not given in the table.

The proposed system is designed to solve image recognition problems, including overlapping classes tasks, when information is submitted for processing in an online mode. The proposed learning algorithm demonstrates both high conversion rate and additional filtering properties. Carried out experiments proved the proposed system to be efficient in solving emotion recognition tasks as well as its declared simplicity.

**Author Contributions:** Y.B.: conceptualization, formal analysis, methodology, writing—original draft; N.K.: data curation, investigation, validation; O.C.: visualization, writing—review & editing.

**Funding:** This research received no external funding.

**Acknowledgments:** The authors thank the administration of the Kharkov National University of Radio Electronics for the administrative support of the research performed.

**Conflicts of Interest:** The authors declare no conflict of interest.

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
