# Peer review of "The Extended Multidimensional Neo-Fuzzy System and Its Fast Learning in Pattern Recognition Tasks"

_data, 2018_

Reviewer 1 Report

In this paper, the authors propose a system based on extended multidimensional neo-fuzzy units for data streams processing. The authors also discuss a  learning algorithm based on the entropy-information criterion.

The article is interesting and well aligned with the journal scope.

Some comments:

1) There are several typos. I suggest having the manuscript proofread.

2) The introduction should be expanded to include more details on the rationale beyond the solution proposed. Also include a description on the structure of the paper.

3) The quality of the figures should be improved.

4) The discussion on related state-of-the-art research is very limited. This should be included in a separate section and be expanded significantly.

5) The conclusions section is far too short and must be expanded.

Overall this is an interesting paper, but it must be expanded to better justify the research rationale and the conclusions.

Author Response

The authors thank an anonymous reviewer for valuable comments that led to manuscript improving.

We've made the following changes, ansvering Reviewer remarks:

2)  
The introduction should be expanded to include more details on the rationale beyond the solution proposed. Also include a description on the structure of the paper.  

     - The introduction has been expanded and updated. A description of the paper structure is included. It was marked light green.

4) The discussion on related state-of-the-art research is very limited. This should be included in a separate section and be expanded significantly. 

 - The separated section about related works was added. It was marked light green. 24 references were added, they were marked light magenta in reference list.

5) The conclusions section is far too short and must be expanded. 

    - The conclusion section was expanded. The updated fragment was marked light green.

Two remarks were:

1)   There are several typos. I suggest having the manuscript proofread.

     - Sorry, we do not have the manuscript proofread.

3) The quality of the figures should be improved. 

     - Sorry, we had not understand, by what criteria we must improve the image.

Reviewer 2 Report

In this paper the concept of the softmax function is introduced to the neuro fuzzy systems.

The parer cites mainly previous work (9 papers) of the main author as state of the art. This indicates that the work is of little relevance for the scientific community.

The evaluation is very weak. The origin of the dataset is not clear.

Only one experiment was performed on 344 images. No baseline is given. The numbers in the results table can be derived from each other and therefore a duplicate.

The correct separation between training and test data is not clearly described. Especially the term "learning set" is not common and gives room for interpretation.

With such a small data set the reader would expect a n-fold crossvalidation, but the whole numbers in the results indicate that there was only one run.

Because of the weak evaluation the conclusions does not hold.

Author Response

The authors thank an anonymous reviewer for valuable comments that led to manuscript improving.

We've made the following changes in response to the reviewer's comments:

1) The parer cites mainly previous work (9 papers) of the main author as state of the art. This indicates that the work is of little relevance for the scientific community. 

    - The literature review was expanded, the state of the problem is in a separate section. These parts of text marked light green. References list was updated, 24 references were added, they were marked light magenta. Some previous work of main author was excluded, this fragment of References list is marked green.

2) The evaluation is very weak. The origin of the dataset is not clear. 

    - An explanation of how dataset from a known database was formed is given. This fragment of text is marked light cyan.

3) Only one experiment was performed on 344 images. No baseline is given. 

     - The numbers in the results table can be derived from each other and therefore a duplicate. Duplicate table rows were excluded.

4) The correct separation between training and test data is not clearly described. Especially the term "learning set" is not common and gives room for interpretation. 

     - Explanation how the data was divided into learning and testing was given. Expression "learning set" was replaced by “Learning data”. The separation description was added. This fragment was marked yellow.

5) With such a small data set the reader would expect a n-fold crossvalidation, but the whole numbers in the results indicate that there was only one run. 

    - The number of terms in nonlinear synapses, the number of neo-fuzzy neurons in input layer and the learning epochs number were changed in every experiment. It was more than 30 runs. But not all the results were included into the Table 1.

6) Because of the weak evaluation the conclusions does not hold. 

    - The conclusion section was expanded. The updated fragment was marked light green.

Reviewer 3 Report

The paper proposes a system based on extended multidimensional neo-fuzzy units and its learning algorithm that meets the conditions for data streams processing. The learning algorithm is based on the use of entropy-information criterion, has significantly improved approximating properties of the system. I think the topic is interest. The proposed method is novel. This paper is well written. It can be accepted after improving the language.

Author Response

The authors thank an anonymous reviewer for comments.

Round  2

Reviewer 1 Report

I am happy with the changes and I recommend pubblication

Author Response

The authors thank the reviewer for valuable comments. The figures for the article were made anew, with a higher resolution. Figures in the text are not marked with color - both were changed.
